# MODELING COMPLEX VECTOR DRAWINGS WITH STROKE CLOUDS

**Alexander Ashcroft**[1]    **Ayan Das**[1]    **Yulia Gryaditskaya**[1,2]    **Zhiyu Qu**[1]    **Yi-Zhe Song**[1,2]

[1]SketchX, CVSSP, University of Surrey, UK  [2]Surrey Institute for People-Centred AI (PAI), UK

## ABSTRACT

Vector representations offer scalability, editability, and storage efficiency, making them indispensable for a wide range of digital applications. Yet, generative models for vector drawings remain under-explored, in particular for modeling complex vector drawings. This is in part due to the primarily sequential and auto-regressive nature of existing approaches failing to scale beyond simple drawings. In this paper, we introduce a generative model for *complex* vector drawings, representing them as "stroke clouds" – *sets* of arbitrary cardinality comprised of n-dimensional Bézier curves. Stroke dimensionality is a design choice that allows the model to adapt to different levels of sketch complexity. We learn to encode this *set of strokes* into compact latent codes by a probabilistic reconstruction procedure based on *De-Finetti's Theorem of Exchangeability*. A generative model is then defined over the latent vectors of the encoded stroke clouds. Thus, the resulting "Latent stroke cloud generator (LSG)" captures the distribution of complex vector drawings in an implicit *set space*. We demonstrate the efficacy of our model in the generation of complex Anime line-art drawings.

## 1 INTRODUCTION

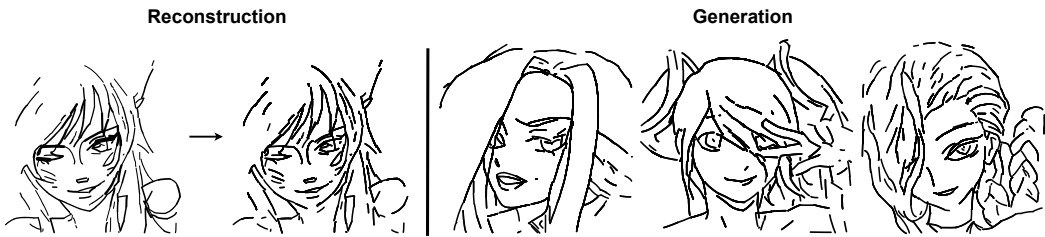

Figure 1: Our model allows us to perform both probabilistic reconstruction of existing samples and generation of new samples matching a training data distribution. **Reconstruction:** We encode a vector drawing (left) as a *set* of Bézier curves and then probabilistically decode it with an MLP-based diffusion model to recreate the drawing (right). **Generation:** Through the use of a latent diffusion model we generate latent codes, and then decode them into vector drawings.

The rise of Diffusion Probabilistic Models (DPMs) (Sohl-Dickstein et al., 2015; Ho et al., 2020a) and their spectacular performance on conditional image generation (Rombach et al., 2022; Saharia et al., 2022) has prompted the emergence of the subfield of creative visual intelligence. However, diffusion-based generation targets raster images and leaves drawings, sketches, and other forms of "chirographic data"[1] underrepresented. Vector representations are scalable, editable, and storage efficient – properties that are beneficial in digital use cases. Such vector representations, however, due to their variable size nature, are mostly (with notable exceptions, *e.g.* Jain et al. (2023)) incompatible with the current image generation pipelines. It is to be noted that modeling chirographic vector modalities have been attempted before (Ha & Eck, 2018; Carlier et al., 2020; Aksan et al., 2020;

---

[1]The term coined by Das et al. (2022; 2023)

Ribeiro et al., 2020; Lopes et al., 2019), but at a smaller scale and complexity. In this paper, we attempt to design a generative model directly on vector drawings, specifically some *complex* ones as depicted in Fig. 1.

Prior attempts at modeling vector drawings were limited to *sequential* representations (Ha & Eck, 2018; Aksan et al., 2020; Ribeiro et al., 2020). The sequence elements used were discretized 2D points (Ha & Eck, 2018; Ribeiro et al., 2020), SVG command tokens (Carlier et al., 2020; Lopes et al., 2019) and individual strokes (Aksan et al., 2020; Das et al., 2020b; Chowdhury et al., 2022). Generative models based on sequential representations were demonstrated only for drawings and sketches significantly simpler in complexity than the ones we target in this work. This can be attributed to the inability of sequential models (*i.e.* RNNs as in Graves (2013), causal Transformers Ribeiro et al. (2020); Aksan et al. (2020)) to handle long sequences. A challenge of modeling long-range dependencies is a well-known drawback (Pascanu et al., 2013), of such models, even beyond chirographic data, in tasks such as scene synthesis (Paschalidou et al., 2021; Para et al., 2023).

In this paper, we target a generative model for vector drawings, specifically focusing on complex drawings. To this end, we abandon the sequential approaches. We propose to model highly complex drawings as a *set* of its constituent strokes – denoting this data structure as "Stroke Cloud". Each stroke can be theoretically represented by any available vector format, such as simple 2D polylines (Ha & Eck (2018)), Bézier curves, or differential geometric embeddings (*e.g.* Aksan et al. (2020)). We here chose Bézier curves for their ubiquitous use in vector art. We then define our generative model on the space of *stroke clouds embeddings (sets embeddings)*, mitigating the long-range dependency issues of sequential representations.

More formally, we define a generative model over a set $\mathcal{X}$ of arbitrary cardinality. Note that generative models are typically defined on the Euclidean vector space $\mathbb{R}^N$ and do not naturally extend to set spaces, which have element invariance properties. However, the foundation for our set-based generative model has been laid by the recent work of Zaheer et al. (2017), and our solution is supported by *De-Finetti's Theorem*. It follows that we can represent a set as a product of independent conditional distributions *over the elements*, given a latent embedding $\mathbf{z} = \mathcal{E}(\mathcal{X})$ of the entire set (Lee et al., 2019; Zaheer et al., 2017). We can then proceed to build a generative model $p_\theta(\mathbf{z})$ over the set embeddings. We show that such models efficiently scale up to complex drawings and generate plausibly-looking samples.

Our main contributions can be summarized as follows: (i) We introduce the first generative model for *complex vector* drawings. Yet, our approach can be used for simpler drawings and any other form of "chirographic data", as we demonstrate in the Appendix. (ii) We define our generative model over complex drawings with a novel "stroke cloud" representation, which is a *set* of its constituent strokes. To this end, we learn *set embeddings* for each set (stroke cloud) to facilitate the downstream generative model.

The code and the data are available at https://github.com/Co-do/Stroke-Cloud.

## 2 RELATED WORKS

**Sketch Generation** First, we focus on the most related recent works for sketch representation and generation. Generating freehand-like sketches from reference images or textual captions remains a challenging task. When generating sketches from images, methods often do not take the abstraction of concepts or objects present in freehand sketches into account, and the produced sketches are some forms of edgemaps (Xie & Tu, 2015; Li et al., 2019; Chan et al., 2022). Alternatively, there is a wide range of sketch generation work based on authentic sketches ranging from creative sketch generation (Ge et al., 2020), shading sketch generation (Li et al., 2020), image-to-sketch translation (Liu et al., 2020) and face sketch synthesis (Wang et al., 2020; Gao et al., 2023). Most of the models are based on a raster sketch representation, which does not reflect the stroke-based nature of authentic drawing that vector sketches do.

To model sketches as a sequence of strokes, sequential and auto-regressive approaches have been used (Ha & Eck, 2018; Zhang et al., 2017). Further, methods for generating vector sketches include SketchHealer (Su et al., 2020), which is a graph-to-sequence approach, SketchPix2Seq (Chen et al., 2017) a CNN-based approach for vector sketches, SketchODE (Das et al., 2022) a neural

ODE based approach and a Bézier curve based approach (Das et al., 2020b). More recently a denoising probabilistic-based method, SketchKnitter (Wang et al., 2022), has demonstrated state-of-the-art performance in generating simple vector sketches. Despite the impressive performance the requirement to represent all training data with a fixed number of points can prove limiting to wider applications of their method. A recent work (Carlier et al., 2020) attempted to *forcefully* impose permutation invariance on stroke-sequences via Hungarian matching (Kuhn, 1955) – a cubic time algorithm (*i.e.* $\mathcal{O}(n^3)$) that is hard to scale beyond low cardinality samples. CLIPascene (Vinker et al., 2022) generates vector sketches from reference images, relying on a per-image optimization approach.

**Point clouds** In this work, rather than representing sketches as a sequence of strokes, we represent a sketch as a set of (unordered) strokes. This problem bears similarity with point clouds generation and representation. Early point cloud works (Achlioptas et al., 2018; Gadelha et al., 2018) represented point clouds as fixed-sized matrices which enabled existing generative models to be easily applied to the problem. AtlasNet (Groueix et al., 2018) and FoldingNet (Yang et al., 2018) learned a mapping from patches in two-dimensional space to three-dimensional space which was able to deform the two-dimensional patches into point cloud shapes. These methods allowed for the generation of point clouds of variable size while also ensuring permutation invariance. An alternative approach is to consider a point cloud as a distribution over three-dimensional points. A range of different approaches utilize likelihood-based methods for generating point clouds of variable size. Thus, PointFlow (Yang et al., 2019) and DFP-Net (Klokov et al., 2020) utilize normalizing flows to model the distribution of points, while PointGrow (Sun et al., 2020) employs an auto-regressive model. More recently, (Luo & Hu, 2021) presented a probabilistic diffusion model-based approach, where they condition a diffusion model on a latent representation of the point cloud and probabilistically reconstruct point clouds. Colored point cloud generation (Wu et al., 2023) builds on the possible use cases for point clouds as it expands the complexity of the information the point cloud can represent. Our *stroke-cloud* approach seeks to leverage the flexibility provided by recent point cloud works but focuses on how to model more complex elements than 2D/3D point elements. Namely, we focus on how to model vector sketches consisting of a large number of strokes of diverse shapes.

## 3 METHOD

Our generative method consists of two modules: (i) the *Stroke cloud Representation Module (SRM)*, comprised of a Set Transformer Lee et al. (2019) as an encoder and a conditional MLP-based diffusion model as the decoder, and (ii) the *Latent Stroke cloud Generator (LSG)*. The SRM module serves as an encoder-decoder, and combined with the LSG module it allows us to generate new drawings representative of the training dataset. The latent code generated by the LSG is decoded by the SRM into $N$ individual strokes, where $N$ is a hyperparameter. We are therefore able to probabilistically reconstruct complex drawings with a variable number of strokes.

### 3.1 DRAWING REPRESENTATION

We represent a drawing in our dataset $\mathcal{D}$ as a set of strokes $\mathcal{S} = \{\mathbf{s}^{(1)}, \mathbf{s}^{(2)}, \mathbf{s}^{(3)}, \cdots, \mathbf{s}^{(N)}\} \in \mathcal{D}$. Note that the cardinality, $N$, of a set $\mathcal{S}$ (the number of strokes in a line drawing) varies across drawings in the training data.

We represent each stroke in the drawing as a Bézier curve. Unless specified otherwise, we use quadratic Bézier curves, represented as follows: $s^{(1)} = (x_1, y_1, x_2, y_2, x_3, y_3)$, where each pair of $(x_i, y_i)$ are the coordinates of the $i$-th control point. While these relatively simple strokes lack the complexity of authentic hand-drawn strokes they can be used to represent complex drawings. For more information on stroke design and usage of Bézier curves of higher degrees please refer to the Appendix E.

### 3.2 SET REPRESENTATION MODULE

We model our Set Representation Module (SRM) as a generative conditional model. As a generative model, we use a Denoising Diffusion Probabilistic Model (DDPM) (Ho et al., 2020a). The training

objective can be formulated as follows:

$$\max_{\{\Psi\}} \sum_{\mathcal{S} \in \mathcal{D}} \log p_{\{\Psi\}}(\mathcal{S}), \tag{1}$$

where $\Psi$ are trainable parameters.

**Stroke cloud joint probability distribution**    It is challenging to define a generative model directly on the set space $\mathcal{S}$ due to the varying number of strokes in each line drawing. A solution that we explore here is to first learn to transform a set $\mathcal{S}$ into a latent representation. Our solution is inspired by the work by Zaheer et al. (2017) and *De-Finetti's Theorem of Exchangeability*[2] stating that a set (exchangeable sequences of random variables) can be modeled as a probability distribution over its constituent elements given some latent representation of the set.

Therefore, we decompose a drawing $\mathcal{S}$ into conditionally independent parametric density functions of the individual strokes $\mathbf{s} \in \mathcal{S}$, conditioned on a latent embedding $\mathbf{z}$ of a drawing $\mathcal{S}$:

$$p_{\{\theta,\phi\}}(\mathcal{S}) = \prod_{\mathbf{s} \in \mathcal{S}} p_\theta(\mathbf{s}|\mathbf{z} = \mathcal{E}_\phi(\mathcal{S})), \tag{2}$$

where $\mathcal{E}_\phi(\mathcal{S})$ is the drawing encoder into a latent space with learnable parameters $\phi$.

Now, we can consider strokes as I.I.D., and can rewrite the training objective in Eq. (1) as follows:

$$\max_{\{\theta,\phi\}} \sum_{\mathcal{S} \in \mathcal{D}} \log p_{\{\theta,\phi\}}(\mathcal{S}) = \max_{\{\theta,\phi\}} \sum_{\mathcal{S} \in \mathcal{D}} \sum_{\mathbf{s} \in \mathcal{S}} \log p_\theta(\mathbf{s}|\mathbf{z} = \mathcal{E}_\phi(\mathcal{S})). \tag{3}$$

**Training**    As an approximation to the true log-likelihood in Eq. (3), we optimize a noise-estimator model trained on noisy versions of strokes $\mathbf{s}_t$ at diffusion timestep (noise level) $t$:

$$\min_{\{\theta,\phi\}} \mathbb{E}_{\mathcal{S} \in \mathcal{D}} \left[ \mathbb{E}_{\mathbf{s} \in \mathcal{S},\ \epsilon \sim \mathcal{N}(\mathbf{0},\mathbf{I}),\ t \sim \mathcal{U}(1,T)} \left[ ||\epsilon_\theta\left(\mathbf{s}_t, t \mid \mathbf{z} = \mathcal{E}_\phi(\mathcal{S})\right) - \epsilon||_2^2 \right] \right] \tag{4}$$

where $\epsilon_\theta(\cdot)$ is the *conditional* noise-estimator. A noisy stroke, $\mathbf{s}_t$, is obtained as follows: $\mathbf{s}_t = \sqrt{\alpha_t}\mathbf{s} + \sqrt{1 - \alpha_t}\epsilon$ for all timesteps $t \in [1, T]$, where $\alpha_t \in [0, 1]$ is a monotonically decreasing diffusion schedule and $\epsilon \in R^{6 \times 1}$. The noise is added to each control point of a stroke. For more detail on the standard Diffusion Model formulation, please refer to appendix F.

**The stroke cloud encoder $\mathcal{E}_\phi$**    The stroke cloud encoder is an important element of the SRM, as it enables the representation of strokes as independent random variables. Due to its theoretically guaranteed permutation-invariant nature, we use a *Set Transformer* with *Pooling by Multihead Attention (PMA)* (with one seed vector) proposed by Lee et al. (2019) in order to encode a given set $\mathcal{S}$ into a compact latent code $\mathbf{z}$.

**Reconstructing the stroke cloud**    Given a trained noise estimator with the parameters $\theta^*$ and a trained stroke cloud encoder such that $\mathbf{z} = \mathcal{E}_{\phi^*}(\mathcal{S})$, we can decode the set by running any diffusion sampler:

$$\widehat{\mathcal{S}} = \left\{ \widehat{\mathbf{s}}^{(j)} := \text{SAMPLER}(\epsilon_{\theta^*}, \mathbf{z}) \mid j \in [1, N] \right\} \tag{5}$$

where $\text{SAMPLER}(\cdot)$ is any sampling procedure compatible with DDPM training (Ho et al., 2020a). We discuss the choice of the sampler in more detail in Sec. 4.

Note that since we assume the strokes to be independent and identically distributed random variables, the model is not aware of the set's cardinality $N$, which is the number of strokes in a sketch. However, due to the presence of the expectation $\mathbb{E}_{\mathbf{s} \in \mathcal{S}}$ in Eq. (3), the model does implicitly encode the relative importance of each stroke. We treat the cardinality $N$ of the reconstructed set $\widehat{\mathcal{S}}$ as a hyperparameter, and discuss it in more detail in Sec. 4. In Sec. 4, we show that both the hyperparameter $N$ and the exact sampling procedure influence the visual quality of the reconstructed drawing.

### 3.3    LATENT STROKE CLOUD GENERATOR

To enable unconditional generation, we leverage a latent generative model that we term "Latent Stroke cloud Generator" (LSG). To train the LSG model, we extract embeddings of the drawings in our dataset $\mathcal{D}$:

$$\bar{\mathcal{D}} = \left\{ \mathbf{z} := \mathcal{E}_{\phi^*}(\mathcal{S}) \mid \mathcal{S} \in \mathcal{D} \right\} \tag{6}$$

---

[2]We provide more detail on De-Finetti's Theorem of Exchangeability in the Appendix G.

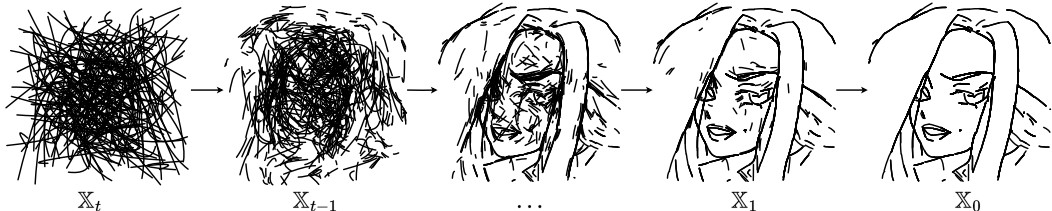

Figure 2: An illustration of the reverse diffusion process for a set of 1000 strokes with a DDIM sampling method. The original drawing was comprised of 350 strokes. Repeated strokes are 're-drawn' on top of one another.

LSG is then a simple generative model defined over the latent vectors $\mathbf{z}$: $p_\psi(\mathbf{z})$ with trainable parameters $\psi$. Just like in Sec. 3.2, we realize $p_\psi(\mathbf{z})$ using a diffusion model. Specifically, we train a parametric noise estimator $\epsilon_\psi$ on the noisy latents $\mathbf{z}_t = \sqrt{\alpha_t}\mathbf{z} + \sqrt{1 - \alpha_t}\epsilon$. This estimator estimates the noise component $\epsilon$:

$$\min_\psi \mathbb{E}_{\mathbf{z}\in\bar{\mathcal{D}},\ \epsilon\sim\mathcal{N}(\mathbf{0},\mathbf{I}),\ t\sim\mathcal{U}(1,T)}\left[\ ||\epsilon_\psi\left(\mathbf{z}_t, t\right) - \epsilon||_2^2\ \right]. \tag{7}$$

## 4 EXPERIMENTS & RESULTS

### 4.1 DATASET PREPARATION

A key challenge in generative modeling for complex vector drawings is acquiring a sufficiently large dataset. Due to the unavailability of such datasets, much of chirographic modeling (Ha & Eck, 2018; Aksan et al., 2020; Das et al., 2023; Wang et al., 2022; Das et al., 2022) has focused on the *QuickDraw* dataset, which contains a vast number of very simple drawings. For reference, the average number of strokes in sketches from *QuickDraw* is 5.

To demonstrate the effectiveness of our stroke cloud-based sketch generation framework in generating complex vector sketches, we synthetically generate a new dataset that we name *Anime-Vec10k*, derived from the *Danbooru2019* dataset (Branwen et al., 2019) of anime raster images. We then use this dataset to train our model.

The *Danbooru2019* image database comprises 3.69 million anime-style artworks in raster format, along with over 106 million annotations. To create our dataset, we randomly select 10,000 samples from a subset of *Danbooru 2019 Portraits*, which are portraits cropped from the original Danbooru dataset. We then transform these samples into line drawings using a style-transfer Generative Adversarial Network (GAN) as described in Chan et al. (2022). Finally, we utilize a line art vectorizer by Mo et al. (2021) to convert these synthetic line drawings into complex vector sketches, consisting of quadratic B'ezier curves. This process is illustrated in Fig. 3. For more details, please refer to Appendix B. On average, the sketches in our dataset consist of 305 strokes.

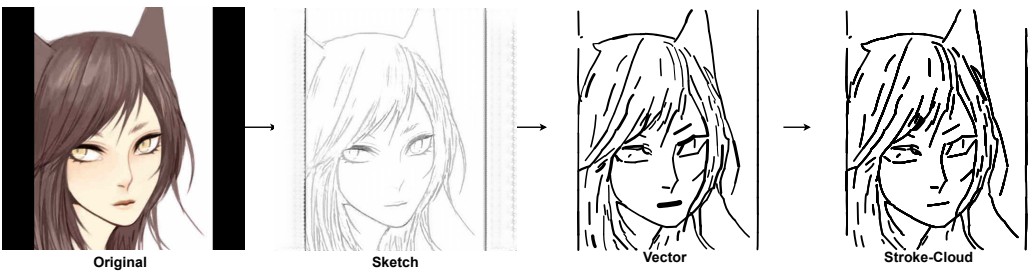

Figure 3: To generate the Anime-Vec10k dataset we take an original image from the Danbooru 2019 Portrait dataset and use a style GAN to convert it to 'sketch style'. We then apply a vectorizer to generate a set of quadratic Bezier curves.

| | Sampler and stroke number | | | | |
|---|---|---|---|---|---|
| | DDIM 100 | DDIM 500 | DDIM 1000 | DDIM 5000 | DDPM 1000 |
| FID | 191 | 34 | 9.8 | 11.3 | 58 |

Table 1: Quantitative comparison of drawings generated under different sampling conditions. DDIM sampling was done with 30 steps while the DDPM sampling was done with 1000 steps.

## 4.2 STROKES REPRESENTATION AND EMBEDDING

As we introduced in Sec. 3.1, in our framework, each stroke in a drawing is represented as $\mathbf{s}^{(i)} = (x_1, y_1, x_2, y_2, x_3, y_3)$, where each pair of $x_i, y_i$ denotes the control points of a quadratic Bézier curve. Using a Set Transformer (Lee et al., 2019), we encode each drawing as a set of variable cardinality, with each stroke as an element in the set. The resulting latent code conditions an MLP-based diffusion model, which learns to generate elements of the set.

To address spectral bias in the MLP (Tancik et al., 2020), we employ a sinusoidal embedding for each control point coordinate. Increasing the dimensionality of the sinusoidal embedding helps to mitigate the spectral bias, as can be seen in Fig. 4.

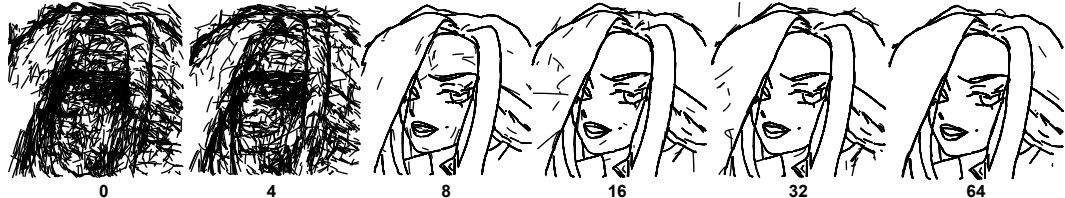

Figure 4: Varying the dimensionality of the sinusoidal positional embedding can have a significant impact on the drawing quality. While the general position of each stroke in the low-dimensional embedding drawings is correct they lack the fine-grained accuracy of the drawings with the higher-dimensional embedding.

## 4.3 STROKES SAMPLING

The variability in the number of strokes within drawings poses challenges in vector drawing generation. While existing methods often employ autoregressive techniques or fix the number of strokes or polyline points, our approach utilizes a more flexible *probabilistic reconstruction process*. This approach allows us to learn compact latent representations while effectively reconstructing these latent codes as observable strokes.

After training, we can condition the SRM with a given latent code to generate samples and attempt to reconstruct the encoded drawing. However, the complexity of the drawing is generally unknown, introducing the challenges of both over-sampling and under-sampling. For more detailed information on probabilistic reconstruction, please refer to Appendix D.

**Over-sampling:** When we generate a significantly larger number of samples than the original number of strokes in the drawing, over-sampling can occur. This is illustrated in the leftmost drawing of Fig. 5. The generative process may result in particular strokes being sampled more frequently, leading to slight variations and noise in some sections of the drawing. Overall, the drawing quality remains largely unchanged, with most strokes being 'redrawn' on top of one another.

**Under-sampling:** On the other hand, under-sampling involves generating too few strokes, resulting in a sparsely populated canvas, as seen in the right-hand drawings of Fig. 5. Under-sampling significantly impacts the quality of the drawing. Table 1 shows the effect of varying the number of generated strokes on the visual quality of the drawing as measured by the FID. These results confirm that the effect of under-sampling on the visual quality is significantly more severe than over-sampling.

**Sampler:** The choice of sampling method can also influence the quality and style of the drawing. Fig. 6 demonstrates the effects of varying the sampling method. Our reconstructive method may re-sample some strokes multiple times, affecting the variance in each sample and, consequently, the

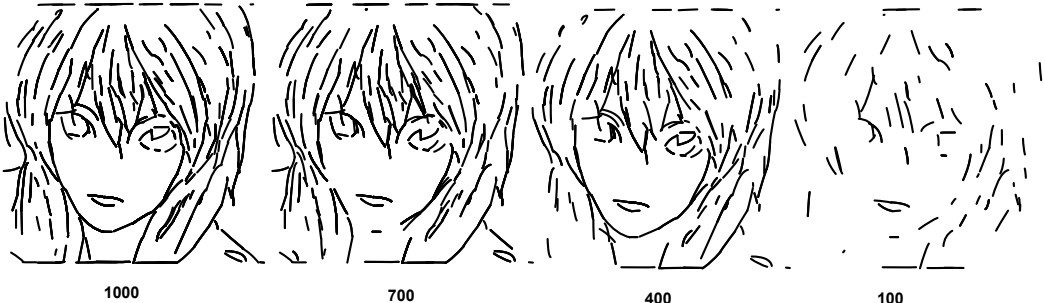

Figure 5: The number of generated samples has a significant impact on the generated drawing. With only 100 samples (right) the drawing is sparsely populated by strokes and key features are missing. However, with 1000 samples (left) a more accurate reconstruction is achieved

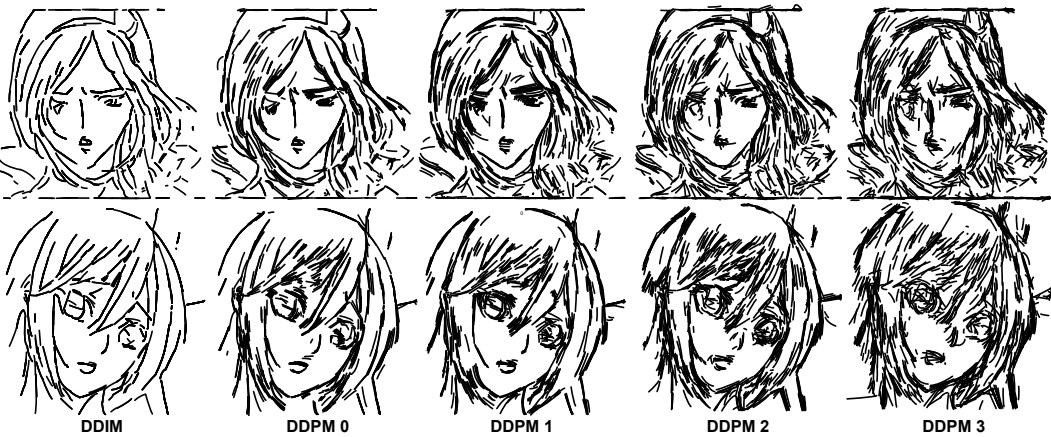

Figure 6: The stochasticity of the sampling method has a significant impact on the quality of the drawings. We utilize a DDIM sampler (left) which is the most resilient to the re-sampling problem. We increase the stochasticity of the sampling process by using a DDPM sampler and multiplying the variance by a scale factor (Das et al., 2023). The scale factor is displayed beneath each drawing.

drawing's appearance. Using a deterministic DDIM sampler produces drawings with clean edges and minimally noticeable re-sampling. The small variance in re-sampled strokes is concealed by the line's thickness. However, using a stochastic sampling method creates a shading effect similar to an artist sketching a composition. To control this effect, we adjust the variance in the de-noising step using a scale factor (Das et al., 2023), increasing or decreasing the stochasticity of the process. Fig. 6 illustrates the impact of this scale factor on the drawing's style.

## 4.4 GENERATION

To generate a drawing we must first generate a latent code with the LSG, this is done using a DDIM sampler and 30 time steps. We decode the resultant latent vector with the SRM, with a DDIM sampler and 30 time steps, into a drawing comprised of 1000 strokes. Drawings generated by our model are shown in Fig. 7. To generate sketches in this figure, we generate the number of strokes much larger than the average number of strokes in the training dataset. While selecting the correct number of samples is an important choice in generating complete and highly complex drawings, if a drawing is incomplete we do have the option of appending more generated samples using the same latent code.

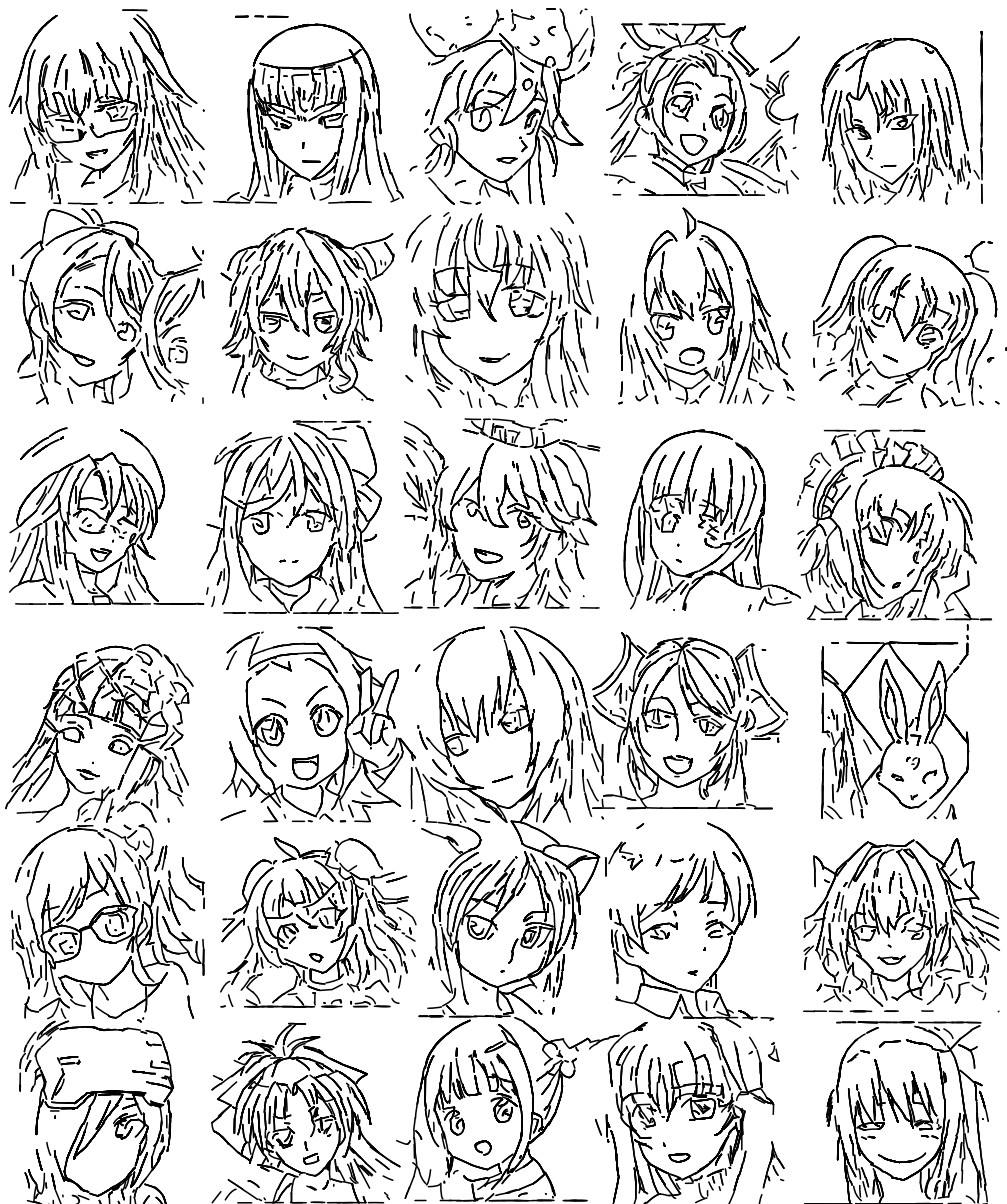

Figure 7: Drawings of 1000 strokes created by decoding latent vectors generated by the LSG with the SRM

## 4.5 INTERPOLATION

The SRM plays a crucial role in reconstructing drawings based on a given latent condition. There-fore, it is essential to assess the model's robustness to different conditions. On the other hand, the LSG's primary function is to generate a latent code that the SRM can decode successfully.

**SRM:** The SRM does not operate unconditionally; instead, it relies on being conditioned by a latent stroke cloud to perform the reconstruction. Consequently, it needs to be resilient to variations in codes generated by the LSG. As illustrated in Fig. 8, when we interpolate between two encoded stroke clouds, the SRM retains semantic features from each stroke cloud, even for conditions it has not encountered previously.

**LSG:** The LSG serves as the core generative component of our model, providing the essential latent code for the SRM to work with. Fig. 9 demonstrates that it is possible to interpolate between two

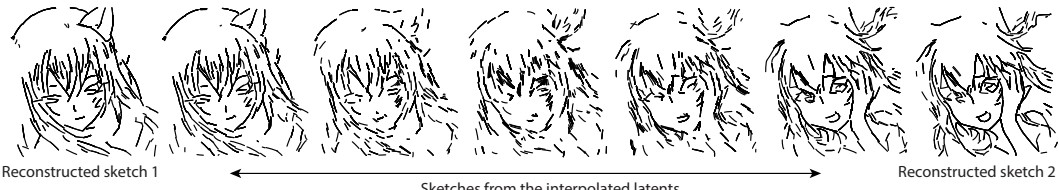

Reconstructed sketch 1        Sketches from the interpolated latents        Reconstructed sketch 2

Figure 8: Interpolating between encoded drawings leads to a gradual morphing between drawings. While noisy in the middle of the transition, clear features are still distinguishable.

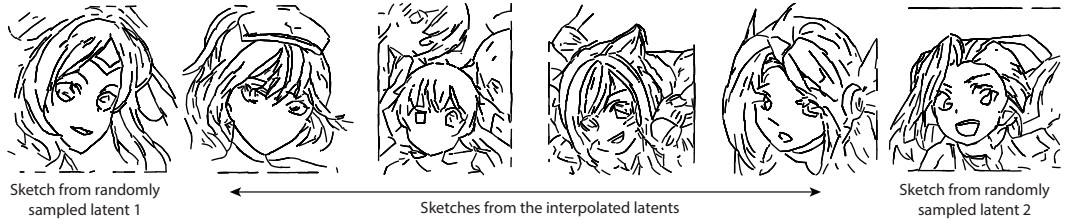

Sketch from randomly sampled latent 1     Sketches from the interpolated latents     Sketch from randomly sampled latent 2

Figure 9: Two random vectors were generated and interpolated between. Each vector was used as the initial vector in the LSG denoising process.

randomly selected noisy vectors, resulting in distinct, drawings. Furthermore, we show in Fig. 10 the impact of gradually adding noise to the originally selected random vector. The sketches obtained from the noised latent vectors are also distinct and plausible.

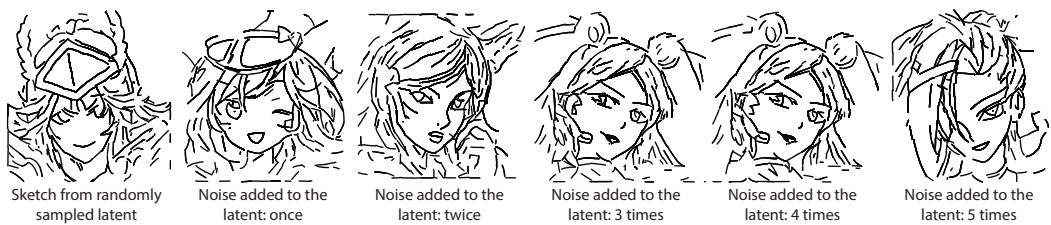

Sketch from randomly sampled latent   Noise added to the latent: once   Noise added to the latent: twice   Noise added to the latent: 3 times   Noise added to the latent: 4 times   Noise added to the latent: 5 times

Figure 10: A random latent vector was sampled, and then Gaussian noise was added gradually to obtain a new latent. The amount of noise added increases from left to right.

## 5 DISCUSSION

In this paper, we propose modeling complex vector drawings as sets of structurally complex elements. We learn to embed these drawings into compact latent codes. These latent codes then condition an MLP-based diffusion model that enables the efficient generation of highly complex vector drawings through a latent diffusion process, supported by De-Finetti's Theorem of Exchangeability. One limitation of our approach is the unknown a priori number of strokes to sample. However, we have shown that oversampling produces visually pleasing sketches in which some strokes overlap. Such strokes can be potentially removed in post-processing by analyzing the areas of overlaps. Limited by the lack of datasets of complex vector drawings, we trained on synthetic data. However, the strokes produced by the automatic vectorizer are shorter than those of hand-drawn sketches. In the supplementary, we provide additional results, showing how our method can support more complex strokes by increasing the number of control points in our stroke representation. Moreover, we also show in the supplementary how additional attributes such as stroke width can be supported by our framework. In summary, we proposed the first approach to model complex vector drawing in a generative context. Our code and the data are available at https://github.com/Co-do/Stroke-Cloud.

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

## A    IMPLEMENTATION DETAILS

**Network.** The SRM network contains a Set Transformer Lee et al. (2019) as an encoder and an MLP-based diffusion model as a decoder. The network architecture can be seen in Fig. 11. Our network was trained end-to-end before the LSG was trained on the resulting latent codes.

**Anime-Vec10k.** A 10k subset of the Anime-Vec10k data set was used for training. The model was trained for 72 hours on a single RTX 4090 with a batch size of 128 and an initial learning rate of 1e-4 that was decayed to 5e-5. After the initial training period, we applied KL annealing for another 24 hours, increasing the KL scale factor from 0 to 1e-8. We trained our model with a linear noise schedule of $\beta_{min} = 1e - 4, \beta_{max} = 1e - 5$ and 200 time steps.

**LSG.** Each LSG was trained on the latent data obtained by passing the training data through the trained encoder. The LSG was then trained for 12 hours with a batch size of 2048 and an initial learning rate of 1e-4 that was decayed to 5e-5. The LSG was trained on a non-conditional version of our MLP. We used a scaled-linear noise schedule of $\beta_{min} = 2e - 2, \beta_{max} = 1e - 4$ and 4000 time steps.

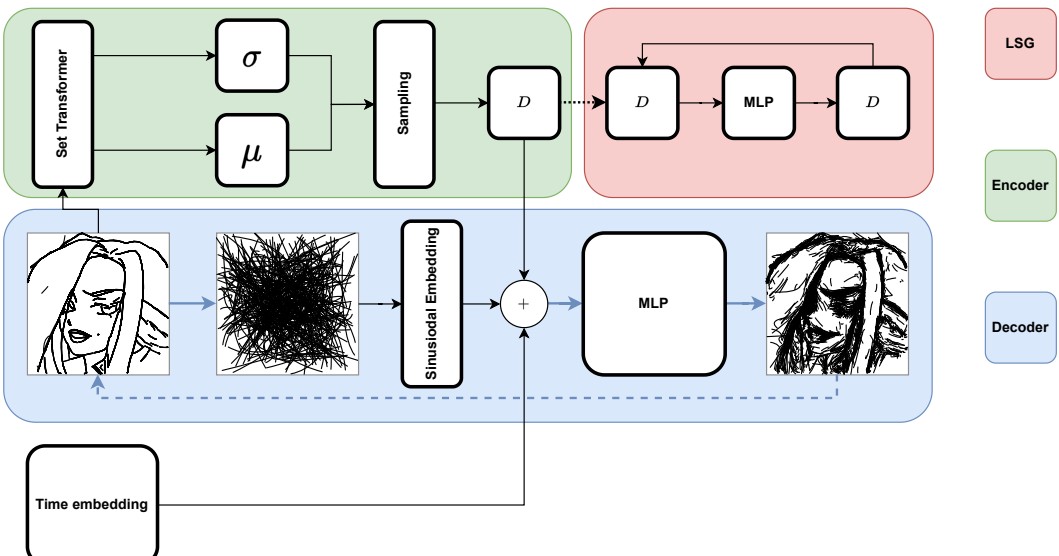

Figure 11: Network diagram of the SRM and LSG for training.

## B    DATASET

Danbooru2019 (Branwen et al., 2019) is a scraping based dataset from the image board Danbooru. Images are scraped and then filtered according to user-generated tags and then subsets of this data are made. The Danbooru2019 dataset has permission from host of the Danbooru website, copyrighted material may have been uploaded by users. Furthermore, the user-generated tags used to filter the images may not be entirely accurate resulting in unwanted images in the final dataset. For our Anime-Vec10k dataset, we manually reviewed all images and removed erroneous content to help address this issue. The examples of training data are shown in Fig. 12.

## C    SAMPLES

In this section, we show more samples from the unconditional generation of the LSG for the Anime-Vec10k model shown in Fig. 14.

Additional interpolation results from the SRM are shown in Fig. 13. In this case, we linearly interpolated between encoded drawings and then used these latent vectors to condition the SRM. Despite

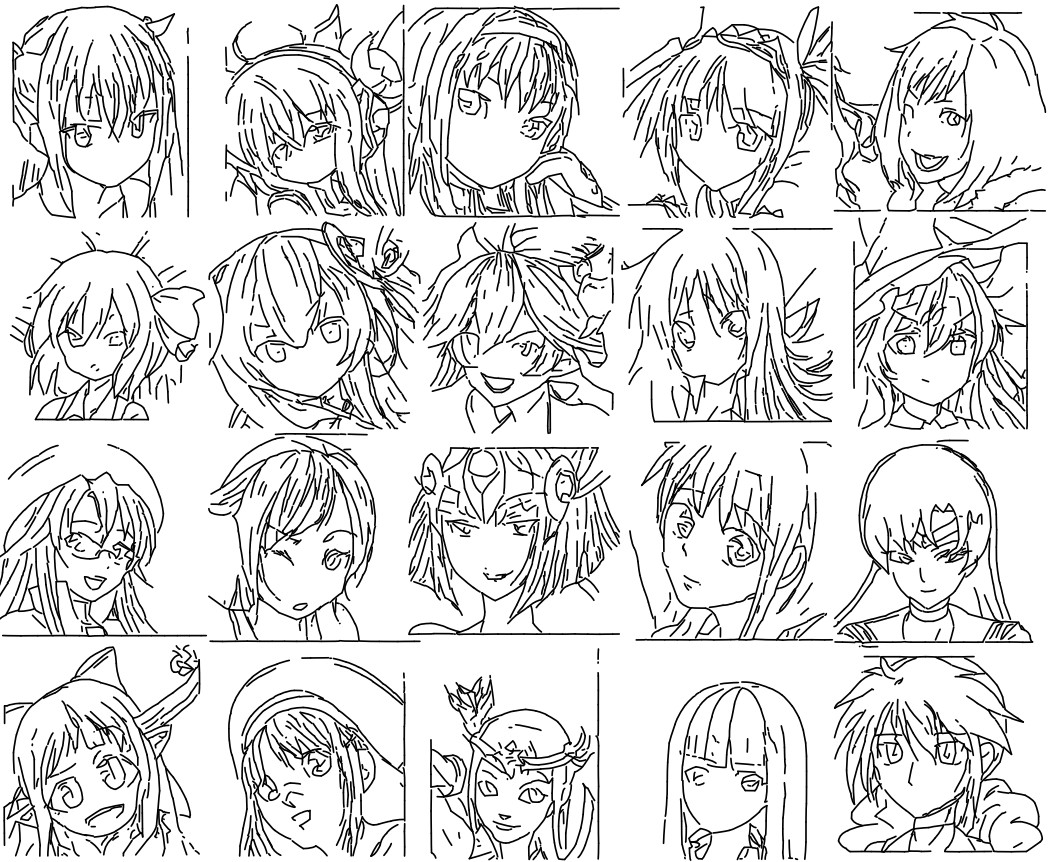

Figure 12: Examples of the training data.

training the SRM with regularization the model is unable to handle conditions that differ greatly from the training data. We consider this to be a limitation of the limited-size dataset.

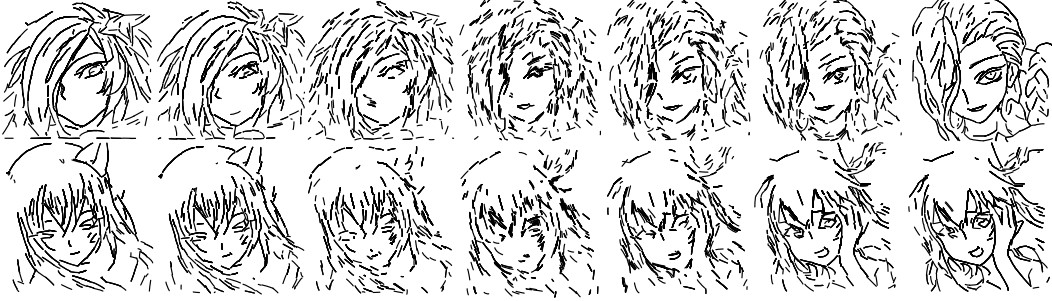

Figure 13: Interpolating between encoded drawings leads to a gradual morphing between drawings. While noisy in the middle of the transition clear features are still distinguishable.

## D    RECONSTRUCTION

This section extends Sec. 4.3 of the main paper. With the probabilistic reconstruction of the drawing, it is unlikely to guess in advance on the *exact* number of strokes required to accurately model it. Instead, it is much more likely that there will be either too many or too few strokes. Fig. 5 showed that selecting too few elements is worse than too many, in addition, Fig. 15 shows that in simpler sketches if too few strokes are generated the incompleteness can be more pronounced. When an

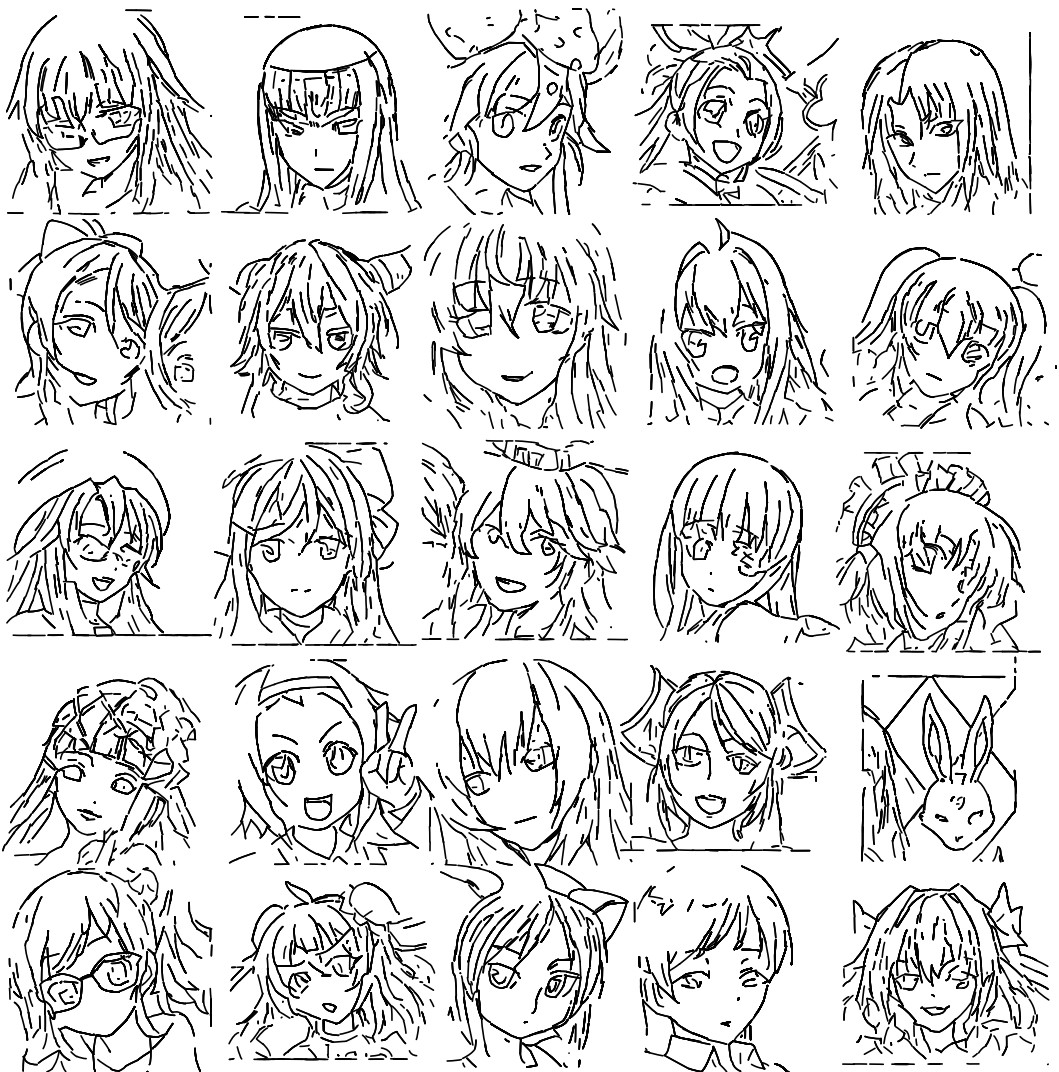

Figure 14: Additional samples generated by our model trained on the Anime-Vec10k dataset.

element is sampled several times there is some variation in that element as was shown in Fig. 6. However, the thickness of each line in the drawing can mask this to make it look like there is no variation present at all. We show more clearly in Fig. 16 that even with a DDIM sampler there is some variation in repeated elements even if they are very similar.

# E STROKE DESIGN

While our method can generate large sets of variable cardinality the visual quality of the results also depends on the complexity and quality of each element in the set: the stroke. We consider two primary limitations in the stroke quality, the availability of training data and the complexity of the stroke the SRM can generate. The line art vectorizer (Mo et al., 2021) formed the backbone of our dataset preparation pipeline. As it produces short line segments in the form of quadratic Bézier curves, we chose quadratic Bézier curves for our experiments in the main document.

However, the general form of each element of the set could be written as follows: $s^{(1)} = (\underbrace{x_1, y_1, x_2, y_2, ....x_n, y_n}_{\text{Control points}}, \underbrace{m_1...m_n}_{\text{Meta}})$, where the control points specify the structural complexity of

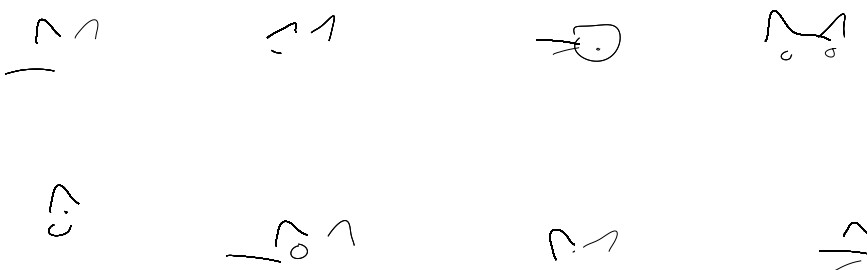

Figure 15: 10 Stroke drawings of QuickDraw cats. The incompleteness of the drawing is quite obvious in simpler sketches.

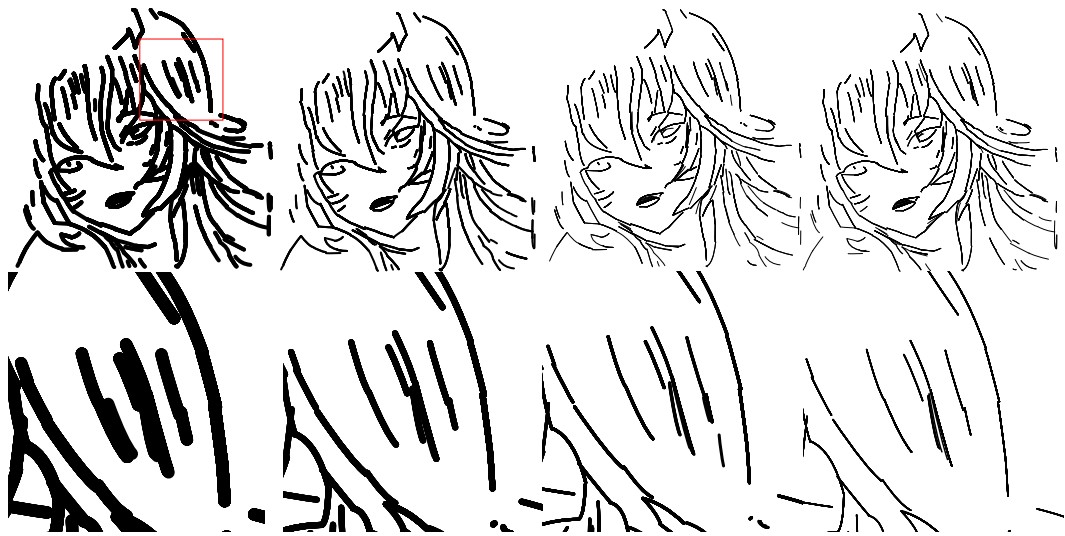

Figure 16: 1000 stroke drawings sampled with DDIM. With thinner rendered lines it is easier to see the variation in stroke placement. Repeatedly sampled strokes are not sampled directly on top of one another but close enough that with thicker lines it is much harder to see.

each stroke, and the Meta-data can include rendering information. Two examples of different stroke designs to what was used in the majority of the experiments can be seen in Figs. 17 and 18.

First, we trained a model on a subset the Anime10k data set but using variable line thickness. The line thickness is provided by the vectorizer we used to generate the dataset (Mo et al., 2021). Each stroke in the set is a 7-dimensional vector $s = (x_1, y_1, x_2, y_2, x_3, y_3, l)$, where $l$ is the line thickness. This serves as an example of how additional metadata can be included in our vector-sketch generative framework. We show the representative, generated by our model, sketches with varying thickness strokes in Fig. 17.

Then, we experiment with a dataset of hand-drawn sketches – the *QuickDraw* dataset. The strokes in this dataset are stored as polylines, consisting of multiple points. We represent cats from the *QuickDraw* with 15 control point Bezier curves (Das et al., 2020a) and then represent each drawing as a set of these strokes. We then trained our model as we did for the Anime10k dataset, with the only difference that now we use 15 control point Bezier curves rather than quadratic Bezier curves. In Fig. 18, we show samples generated with our method.

Both cases show that, while we used a simple stroke design (a quadratic Bézier curve with contract stroke width) for the majority of the experiments, our framework allows us to support more complex strokes and additional stroke attributes.

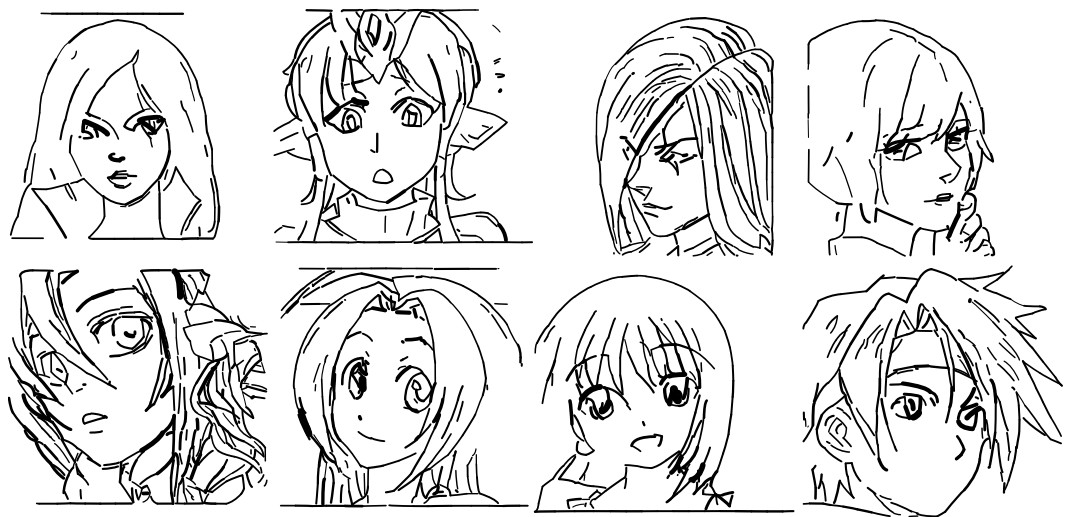

Figure 17: 1000 stroke samples generated with a variable line thickness

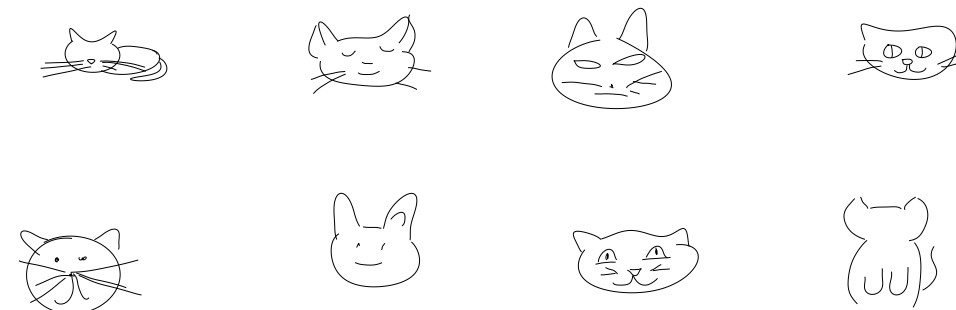

Figure 18: For each sketch example, 50 stroke samples were generated with our method, after training the model on the QuickDraw dataset *cats* category. Each stroke was represented by a 15 control point Bézier curve.

## F  DIFFUSION MODELS BACKGROUND

To learn the probability distribution over data $x$. Diffusion models corrupt training data by slowly injecting noise and then learn to reverse the corruption, such that the obtained models can gradually transform random noise into samples for data generation.

**Forward Process**

For each training sample $x_0 \sim q_{data}(x_0)$, a discrete Markov chain $x_0, x_1, ..., x_T$ is formed by the forward process (also known as diffusion process). This process is defined as a Markov chain which slowly adds Gaussian noise to the data according to a variance schedule $\beta_1, ..., \beta_T$ :

$$q(x_{1:T}|x_0) = \prod_{t=1}^{T} q(x_t|x_{t-1}) \tag{8}$$

$$q(x_t|x_{t-1}) = \mathcal{N}(x_t; \sqrt{1 - \beta_t}x_{t-1}, \beta_t I) \tag{9}$$

If we know $q(x_{t-1}|x_t)$, we could sample data from the data distribution $q(x_0)$ by first sampling $x_T$ from $q(x_T)$ (isotropic Gaussian) and then sampling from $q(x_{t-1}|x_t)$ until we get $x_0$. However, it is difficult to estimate $q(x_{t-1}|x_t)$ since it needs the entire dataset to do so. Therefore, $p_\theta$ is proposed to approximate the conditional probabilities $q(x_{t-1}|x_t)$ during the reverse process.

**Reverse Process** In the reverse process, diffusion models have to denoise the perturbed data (starting at random noise $p(x_T) = \mathcal{N}(x_T; 0, I)$) back to the origin data $x_0$. Mathematically, diffusion models are defined as

$$p_\theta(x_0) = \int p_\theta(x_{0:T}) dx_{1:T} \tag{10}$$

for which the joint probability distribution $p_\theta(x_{0:T})$ defines the reverse process as:

$$p_\theta(\mathbf{x}_{0:T}) = p(\mathbf{x}_T) \prod_{t=1}^{T} p_\theta(\mathbf{x}_{t-1}|\mathbf{x}_t) \tag{11}$$

$$p_\theta(\mathbf{x}_{t-1}|\mathbf{x}_t) = \mathcal{N}(\mathbf{x}_{t-1}; \boldsymbol{\mu}_\theta(\mathbf{x}_t, t), \boldsymbol{\Sigma}_\theta(\mathbf{x}_t, t)) \tag{12}$$

The training object is then to optimize the variational bound on the negative log-likelihood as:

$$
\begin{aligned}
-\log p_\theta(\mathbf{x}_0) &\leq -\log p_\theta(\mathbf{x}_0) + D_{\text{KL}}(q(\mathbf{x}_{1:T}|\mathbf{x}_0)\|p_\theta(\mathbf{x}_{1:T}|\mathbf{x}_0)) \\
&= -\log p_\theta(\mathbf{x}_0) + \mathbb{E}_{\mathbf{x}_{1:T}\sim q(\mathbf{x}_{1:T}|\mathbf{x}_0)}\Big[\log \frac{q(\mathbf{x}_{1:T}|\mathbf{x}_0)}{p_\theta(\mathbf{x}_{0:T})/p_\theta(\mathbf{x}_0)}\Big] \\
&= -\log p_\theta(\mathbf{x}_0) + \mathbb{E}_q\Big[\log \frac{q(\mathbf{x}_{1:T}|\mathbf{x}_0)}{p_\theta(\mathbf{x}_{0:T})} + \log p_\theta(\mathbf{x}_0)\Big] \\
&= \mathbb{E}_q\Big[\log \frac{q(\mathbf{x}_{1:T}|\mathbf{x}_0)}{p_\theta(\mathbf{x}_{0:T})}\Big]
\end{aligned}
\tag{13}
$$

$$\text{Let } L_{\text{VLB}} = \mathbb{E}_{q(\mathbf{x}_{0:T})}\Big[\log \frac{q(\mathbf{x}_{1:T}|\mathbf{x}_0)}{p_\theta(\mathbf{x}_{0:T})}\Big] \geq -\mathbb{E}_{q(\mathbf{x}_0)} \log p_\theta(\mathbf{x}_0)$$

This is the equivalent of:

$$
\begin{aligned}
L_{\text{VLB}} &= L_T + L_{T-1} + \cdots + L_0 \\
\text{where } L_T a &= D_{\text{KL}}(q(\mathbf{x}_T|\mathbf{x}_0) \,\|\, p_\theta(\mathbf{x}_T)) \\
L_t &= D_{\text{KL}}(q(\mathbf{x}_t|\mathbf{x}_{t+1}, \mathbf{x}_0) \,\|\, p_\theta(\mathbf{x}_t|\mathbf{x}_{t+1})) \text{ for} 1 \leq t \leq T - 1 \\
L_0 &= -\log p_\theta(\mathbf{x}_0|\mathbf{x}_1)
\end{aligned}
\tag{14}
$$

The training objective for Eq 12 is to get $\mu_\theta(x_t, t)$, while not involving $\sum_\theta(x_t, t)$, as it is set to time-dependent constants $\sigma_{t=1}^2$. Furthermore, instead of predicting $\mu_\theta(x_t, t)$ by a neural network, Ho et al. (2020b) proposed to utilize an approximator $\epsilon_\theta(x_t, t)$ to predict noise $\epsilon$ from $x_t$, which is proven to be more effective than optimizing $\mu_\theta(x_t, t)$. The simplified training objective is:

$$L_{simple}(\theta) = \mathbb{E}_{t\sim[1,T], x_0\sim q(x_0), \epsilon\sim\mathcal{N}(0,1)}[||\epsilon - \epsilon_\theta(x_t, t)||^2] \tag{15}$$

**Sampling** Once trained, the network estimates noise $\epsilon$ from the sample $x_t$ at timestep $t$ ($\epsilon_\theta(x_t, t)$). $\mu_\theta(x_t, t)$ can then be derived from $\epsilon_\theta(x_t, t)$ by the following:

$$\mu_\theta(x_t, t) = \frac{1}{\sqrt{\alpha_t}}(x_t - \frac{1 - \alpha_t}{1 - \overline{\alpha}_t}\epsilon_\theta(x_t, t)) \tag{16}$$

where $\alpha_t = 1 - \beta_t$ and $\overline{\alpha} = \prod_{s=1} \alpha_s$. We can then sample data from $p_\theta(x_{t-1}|x_t)$ according to Eq 12 until we reach $s_0$

## G  DE FINETTI'S THEOREM

**Exchangeability**

For a given set $\{X_i\}_{i=1}^n$ of objects let $\mu_{X_1...,X_n}$ denote the joint probability. This set is exchangeable if $\mu_{X_1...,X_n} = \mu_{X_{\pi(1)}...,X_{\pi(n)}}$, *i.e.* if for every permutation $\pi : \{1, ..., n\} \rightarrow \{1, ..., n\}$.

**De Finetti General Representation Theorem**

If $X_1, X_2, ..., X_n$ is an exchangeable sequence of variable length with probability measure $P$, then there exists a distribution function $Q$ on $F$, the set of all distribution functions on $\mathbb{R}$, such that the joint distribution of $(X_1, X_2, ..., X_n)$ has the form:

$$p(X_1, X_2, ..., X_n) = \int_F \prod_{i=1}^n F(X_i) dQ(F) \tag{17}$$

where $F$ is an unobservable distribution function.

$$Q(F) = \lim_{n \to \infty} P_n(\hat{F}_n) \tag{18}$$

is the probability measure on the space of functions $F$, defined as a limiting measure $n \to \infty$ on the empirical distribution function $\hat{F}_n$

