# OpenReview forum: "Modelling complex vector drawings with stroke-clouds"
_ICLR.cc/2024/Conference — ICLR 2024 poster_

### Official Review · Reviewer_LS4E · 2023-10-30

**Soundness:** 3 good
**Presentation:** 3 good
**Contribution:** 2 fair
**Rating:** 6
**Confidence:** 5

**Summary:**

The idea is to model a vector drawing as a _set_ of strokes.
Drawings are created by a diffusion process to create a “stroke cloud”.
The strokes are represented in a low-dimensional latent space, making them conditionally independent.
The paper tackles an interesting problem in a novel and principled way.
It produces reasonable results.
It is of publishable quality.

**Strengths:**

Introduction:
Differentiates between vector and pixel representations, and claims the former is pressed (somehow)
on creativity while the latter is based on perception. These claims I have weak sympathy with, but
which I don’t fully agree with (surely people draw with lines/vector because that is how they perceive
the world). The are made without any citation into either the perceptual or artistic / art-history literature.
And they pervade the first part of the paper, making slightly uncomfortable reading for me - feeling that
the argument made may even mislead. I’d like to see it changed, but given my greater sympathy towards
the use of vectors over pixels, it’s not something I will insist upon other than “think again about the argument”.

Related Works:
The paper touches upon only a very small fraction of the available literature, and fails to cite any paper prior to 2015 - even though there is a large body of work (see the non-photorealistic rendering literature).
More on this later.

Method
The method is to represent a drawing with a set of strokes, the exact number does not matter.
I very much like the fact the authors acknowledge strokes are not independent, but use
De-Finetti’s Theorem of Exchangeability - which I am not familiar with (so thanks for the introduction)
so that they are conditionally independent.

I am less convinced by the use of quadratic curves to represent strokes. Strokes can be long
and very complex. Example - a drawing of curly hair may use long, looped strokes. And strokes vary
in width, media density and much more. I understand this is a first step - but some text that acknowledges
the very severe limitations the method operates under is, i think necessary so that the contribution can
be more fairly understood.

It’s not clear to me why the generator produces a face (in the same view angle, or its mirror, and
with the same crop window). This again is very limiting.

Results:
The authors make no experiments at all. This is a significant weakness in the paper.
Rather they show some images that have been generated, and then claim the images look good.
Unfortunately for the authors, others (like me) have a different position.
I am not convinced the drawings are “high quality” as claimed.
Especially when compared to the NPR work my view is that output produced in recent years is low quality.
And no work I have seen in comparable in quality to human art.

The fact drawing are usually made of some actual thing, like a face or a dog.
The paper never mentions how to control the output to direct it to a particular noun class.
(Does the noun class impact the embedding into latent space? I guess so - it depends on S)
All the images are of faces, and all in “manga” style.
This means the paper is not clear on its generality.
Do you have examples of other noun classes?

Conclusion:
I found the conclusion rather limited. I certainly disagree with the claim that
  “the primary limitation lies in the probabilistic nature of the reconstruction process”.
In fact the primary limitation is much more likely to be that the system makes no use of
semantic information, other than possibly implicitly via training. It is this issue, rather than any
other, that has constrained progress in this area.

Summary:
I enjoyed the paper, in part because of the controversial claims it makes, but mostly because it
takes on a difficult problem in an imaginative way. That said, I feel the authors would do themselves and
the field much great justice if they were to resist their claims. I strongly recommend looking at some
real drawings, ideally “in the flesh” - and not just manga images of 3/4 faces, heavily cropped, drawn
with short quadratic curves of uniform width and density, on a perfectly flat surface.

The paper lacks any experiment, which was once common but is far less so now. One obvious
experiment is to conduct some kind of Turing test, of which there are many variants in the NPR/NST
literature (especially the more recent). As it stands the paper lack rigour - a rigour which I suspect would
lead the authors to question some of their more controversial (for me) claims.

The paper makes a contribution and is publishable.

**Weaknesses:**

Please see above.

**Questions:**

How well do you you expect the system to generalise to?
* other objects
* other points of view
* other styles
* the true complexity of real strokes

What citable evidence do you have that vectors relate to creativity and pixels to perception?

Why did you conduct no experiment?

Why have you not cited any paper prior to 2015, when there is plenty of relevant work?

What is your defence fot the claim that "the primary limitation lies in the probabilistic nature of the reconstruction process"?

---

> ### Author Response · Authors · 2023-11-23
>
> Thank you for your detailed comments!
>
> **(R3-1) Introduction**
> Thanks, we have now revised the introduction to remove subjective claims. Our main point is that the vector format is not only a scalable alternative for a raster format, but it also allows us to model additional information lost in the raster representation: that of each drawing and sketch being a collection of individual strokes. Such representation enables more fine-grained editing taking the artwork creation process into account. Please do inform us if this is now more accurate than previously stated — it certainly was not our intention to overclaim!
>
> **(R3-2) Additional references**
>
> We focus our discussion on recent papers as the focus of our paper is on deep generative approaches for vector graphics. We are happy to provide a more extensive discussion of other related topics if reviewers think that this will benefit the paper's exposition.
>
> **(R3-3) The use of quadratic curves to represent strokes (strokes complexity)**
>
> First, we would like to stress that, theoretically, SRM has no restriction on the exact stroke model to use. The reason why we used quadratic Bezier curves is that the method used to create vector sketches produces fairly simple strokes, and quadratic Bezier curves are more than enough to model strokes in the training set.
>
> To show that our model supports Bezier curves of a high degree (can represent more complex strokes), we train on the sketches from the QuickDraw dataset with Bezier curves with 15 control points.
> We added Figure 17 to the Appendix which shows that our model can flawlessly support more complex curves.
>
> **(R3-4) Additional stroke parameters
> (Same as R2-3)**
> Thanks, great suggestion. While we leave extensive evaluation of adding additional properties to strokes to future work, as a proof of concept, we experimented with adding line thickness to a set of control point positions, describing each stroke. A newly added Figure 16 in the supplemental shows that our model manages to leverage line thickness as well and represents eyelashes with thicker lines, showcasing flexibility in our approach.
>
> **(R3-5) Results view angles and classes**
>
> The short answer is the results simply reflect the training data. We train only on the dataset of faces. But of course, our method is agnostic to the exact nature of data, as long as they can be categorized as *complex sketches*.
>
> Additional results for a different dataset (QuickDraw) are shown in the newly added Figure 17 in the Appendix.
>
> **(R3-6) Generating a sketch of a particular noun class**
>
> Enabling more classes is possible in a naive setting by sourcing a similar dataset where we can readily obtain vectorised training sketches, and training on millions of these sketches. This was however not possible within the rebuttal period, especially recognising our limited computation resources.  Another option could be conditioning our SRM module on feature spaces of large pretrained models and focusing on the transfer learning problem, this is however out of the scope of our work.
>
> **(R3-7) Numerical evaluation**
>
> Thanks. We have added a comparison of the FID scores of the generated drawings under different sampling conditions in Section 4.2 Table 1.
>
> **Summary**
>
> Given the answers 3-6, we expect our model to generalize well to new classes and styles, with stroke design being an area of study moving forward.

---

### Official Review · Reviewer_FQwM · 2023-10-31

**Soundness:** 2 fair
**Presentation:** 2 fair
**Contribution:** 3 good
**Rating:** 6
**Confidence:** 4

**Summary:**

This paper proposes a method using Diffusion Models and  Set Transformer for generating sketchs of Anime girls in vector format.
The training data is obtained by transfering raster images into vector curves.
The generation module contain two parts, the Stroke cloud Representation Module (SRM) and the Latent Stroke cloud Generator (LSG).
The LSG module is a diffusion model, which serves to sample latent codes. Each latent code corresponds to a sketch result.
The SRM module contain two parts:
The first part is a Set Transformer, which serves as an encorder to encode training datas into latent codes.
The second part is a diffusion model, which serves as an decorder to generate sketch strokes conditioned on the latent code.

**Strengths:**

1. The idea of applying the Set Transformer to encode the stroke data is the most significant advantage of this paper. The nature of the Set Transformer theoretically guarantees permutation-invariant, so that all the strokes in one sketch can be encoded into a latent code to represent the sketch without worrying about the order of the strokes. This encoder also works even if the stroke number varies among the sketches in the dataset.

2. Also thanks to the Set Transformer, the decoder of the SRM module can sample arbitrary numbers of strokes to constitute a sketch.

3.  Although this paper focuses on the topic of sketch generation. I see the potential of this method to be applied to other artistic styles such as oil painting, just needs to change the stroke model and the dataset.

**Weaknesses:**

1. I think the main weakness of this paper is the poor visual quality, which is far from artistic application. And the stroke design of the Bezier curve is too simple.  Could the Bezier curve contain more parameters such as width and transparency?

2. Lack of comparison. I do recognize that there may be no methods similar to your technique route. But you should compare with at least one method related to the topic of sketch generation, even though there may be raster-based such as GANs.

3. There seems to be a flaw in the experiment of section 4.4, figure 8. I can see the sketches corresponding to the interpolated results in Figure 8 (the middle ones) are obviously not in the correct domain. This may be because your interpolation function does not fit this distribution (and I didn't see your interpolation function), which leads to the interpolated latent codes are not on their true distribution（you can analogy the manifold of Swiss Roll).

**Questions:**

This question is just for discussion (doesn't affect my rating):
How do you critique the technique route that transfers raster images into vectors? Just as your method of establishing your dataset.
I mean, people can use raster image generation tools such as Stable Diffusion to generate a raster image, and then transfer it into SVG format (the recent work of VectorFusion follows this idea). Compared with this technique route, what's the advantage of your technique route? Or in other words, do you think directly generating vectors is more prospoective?

---

> ### Author Response · Authors · 2023-11-23
>
> Thank you for the detailed feedback and appreciation of our use of the set transformer.
> We address each question one by one:
>
> **(R2-1)** **The visual quality of the sketches**
> We focus on *complex vector sketch* generation as a new genre where sketches can have a large number of constituent strokes. We make no assumption about the exact nature of sketches in our theory as long as it fits into the said criteria. However, one needs to take into account the fact that there are no large-scale vector datasets of complex sketches available. The limitation on the visual quality of sketches primarily comes from the training data and the quality of the outputs of the available vectorization methods.
>
> **(R2-2) Bezier curves complexity**
> Firstly, we would like to mention that, even though we used quadratic Bezier curves as a design choice, we never made any hard assumptions about the exact stroke representation in our theoretical setup. To further clarify any doubts, we have now added Figure 17 to the appendix, generating sketches from a model trained on the QuickDraw dataset. In this case, we used Bezier curves with 15 control points (able to represent significantly complex strokes). It can be seen that our model copes well with more complex strokes when the appropriate training data is available.
>
> **(R2-3) Additional stroke parameters**
> Thanks, great suggestion. While we leave extensive evaluation of adding additional properties to strokes to future work, as a proof of concept, we experimented with adding line thickness to a set of control point positions, describing each stroke. A newly added Figure 16 in the supplemental shows that our model manages to leverage line thickness as well and represents eyelashes with thicker lines, showcasing flexibility in our set transformer approach.
>
> **(R2-3) Quantitate results**
>
> Thank you for the suggestion. We have added a comparison of the FID scores of the generated drawings under different sampling conditions in section 4.2 Table 1.
>
> **(R2-4) Interpolation results in Figure 8**
> We agree that the interpolated results in Figure 8 (now Figure 10.) are of lower quality. We however argue that this might not be a limitation of our SRM module but is a result of the lack of training data for the latent generative model. This however should not weaken our main contribution which is complex vector sketch generation. We will now move this Figure to the Appendix and provide a careful discussion around it.
>
> **(R2-5) The advantage of our approach of direct vector sketch generation over raster sketch generation followed by a vectorization step.**
>
> This is exactly where our method shines! Our approach is very time-efficient, something we believe is critically important for complex vector sketch generation — it generates a 1000-stroke drawing in just ~0.05 seconds. For comparison, it takes 10-15 seconds to vectorize a raster sketch with the approach we used to prepare the training set. Moreover, being able to generate vector sketches directly has the potential to provide more control over the generation process and editability of the output.

---

### Official Review · Reviewer_DN8s · 2023-10-31

**Soundness:** 2 fair
**Presentation:** 3 good
**Contribution:** 2 fair
**Rating:** 5
**Confidence:** 4

**Summary:**

The paper purports to present a method for generating _vector representation of complex sketches_.
The problem is important and has the following challenges:
1. most work on generative modeling focuses on images, so vector representation is underrepresented.
2. the number of _strokes_ within each sketch is variable.
3. autoregressive methods using a sequence representation fail to scale up.
4. the sequence representation is sensitive to the ordering of strokes.

---

The claimed contributions are:
1. proposing the first method for generating _highly complex drawings_.
2. using a _set representation_ instead of a sequence representation.

---

The paper:
1. (MAJOR) argues that a set representation is is theoretically possible because  of  `De Finetti General Representation Theorem` which states that for an `exchangeable` set with a pdf for each of it's constituent sets (one realization of the permutation), there exists a pdf that describes all the permutations in the limit.
2. (MAJOR) presents a practical way to construct this permutation-invariant pdf, so that we can sample sketches from this distribution tractably.
3. (MINOR) presents a new dataset called `Anime-Vec10k` which which is significanlty more complex than existing datasets (notably QuickDraw!)

---

The method used to achieve the goal of  "generating _vector representation of complex sketches_" is roughly the following:
1. Consolidate image into sets of bezier-curves.
2. Use a Set-Encoder to encode the stroke-set to get a latent.
3. Use a VAE loss to regularize the space of latents.
4. Use an MLP based conditioning denoising network to get from (500/1000) _random_ strokes to the actual strokes.
5. The important point in point 4 is that **each stroke is considered iid**.

**Strengths:**

**The problem setting is valuable to the community.**

---

While diffusion models have been providing very high quality generative models in the raster world, they have not made as much of a splash in the vector world. So I commend the authors for tackling this problem.

**Very good exposition.**

---

The paper is very easy to follow. While the abstract could possibly use some more technical details. Everything from the introduction on flows logically. The introduction is superb, even though it spends a bit more time describing why `creational` representations matter than I would have liked. The introduction still sets up the problem well - the problem with sequences (ordering and non-scalability of the autoregressive approach) and the problems with using a set approach (variable and unknown cardinality). The related work section is quite comprehensive to me. I would still like to suggest two references which might be interesting to the reader in the weaknesses section.

**The core idea is simple to implement**

---

The whole paper is based on simple building blocks, so any (re-)implementation should be easy.

**Weaknesses:**

While I like the paper, in its current form the paper has many weaknesses:

**(MAJOR): Sampling issues**:

---

The authors clearly point out that the number of strokes is an issue, but simply brush it away by saying we use 1000 strokes.
This is a glaring flaw. What if the sketch is simpler? What happens to the other strokes? Are they duplicated?

The discussion in Appendix E is is simply not enough to provide a solution.

**(MAJOR): Matching issues/ iid assumption of strokes **:

---

Related to sampling is the issue of matching. In the diffusion model equation (3), you simply decompose the sketch into iid strokes. This is a **VERY** strong assumption. I do not see how during sampling, one cannot get a degenerate solution of just repetetively denoising into the same stroke. There is no question asked about how the iid sampling affects the proposed pipeline.

In the same way, how is the sequence $\mathbf{s}$ generated at training time? Do you let the randomness of the random strokes take care of matching the proper final stroke? Why is there no hungarian matching? Assuming normalized coordinates in $[-1, 1] \times [1, 1]$ , how does it make sense for the denoising network to take a random stroke at (-1, -1) and try to denoise it to (1,1), instead of just recognizing (with hungarian matching) or simple euclidean distance that it is much easier to denoise the stroke closer to the final location?

**(MAJOR):  Sequence representation/Sequence lengths**

---

It is not clear at a first glance (unless I am wrong) that the paper does the following:
1. Takes in a **variable** number of input strokes
2. Generates (denoises) a **fixed** (1000) number of output strokes.

The motivating factor of the paper, and the way the paper is currently written suggest that the model is capable of generating a variable number of strokes. It seems that is not possible

**(MAJOR) No quantitative results**

---

I said previously that the paper has missing citations. I will mention the actual links later but describe them here:

There has been recent work (<1.5years) in the autoregressive generation area. The work is based on how to make generation permutation invariant which is exactly the problem being tackled here. There are two papers in this area: [Paschalidou] and [Para]. [Paschalidou] use a learnable query vector that looks at the previously generated sequence and predicts the next one. [Para] uses a set encoder and a sequence decoder with mask tokens to perform controled generation.

The authors do not cite these papers. The authors have no baselines as they (rightfully) claim that previous work does not scale - but there [Paschalidou] has code available and it would be easily adaptable to their current training regime

[code](https://github.com/nv-tlabs/atiss)

All you have to do is make each token the sum of its control point embeddings! And then introduce some form of conditioning - either a single condition token or an encoder as done in [Para].

This should significantly strengthen the paper - we see exactly how slow and underwhelming the other methods are, how slow to sample, what the FID is, and qualitative results as well. I would really like to see those result

**(MINOR)  Few qualitative results**

---

While there are a decent number of qualitative results in the paper already - the dataset itslef is small - 10k samples. I would request the authors to **check for overfitting** by also visualziing the closest training set example to each generated sample - this could be in the rgb space as well as some perceptual space - look at the sketch retrieval literature or just VGG features.

Another thing that will help both analyze the dataset quality and the generation quality is to have a big 10x10 grid from both sets (trainig and generation) somehwere in the paper

**Missing citations**

The missing citations are
1. @Inproceedings{Paschalidou2021NEURIPS,
  author = {Despoina Paschalidou and Amlan Kar and Maria Shugrina and Karsten Kreis and Andreas Geiger and Sanja Fidler},
  title = {ATISS: Autoregressive Transformers for Indoor Scene Synthesis},
  booktitle = {Advances in Neural Information Processing Systems (NeurIPS)},
  year = {2021}
}
2. @inproceedings{10.1145/3588432.3591561,
author = {Para, Wamiq Reyaz and Guerrero, Paul and Mitra, Niloy and Wonka, Peter},
title = {COFS: COntrollable Furniture Layout Synthesis},
year = {2023},
series = {SIGGRAPH '23}
}

**Questions:**

I already asked most questions.

1. Figure 5: Why does the DDPM sampler seem to have more strokes than the DDIM sampler? or are the strokes just placed closer together in the DDIM sample?

2. Do you plan to release the dataset?

---

> ### Author Response · Authors · 2023-11-23
>
> Thank you for a very detailed review, *particularly* your appreciation of the challenging and forward-looking nature of the problem we solve, and our idea of using a set representation approach to model vector graphics.
>
> Below we answer each question one by one.
>
> ---
>
> **Strokes sampling during inference**
>
> SRM (Set Reconstruction Module) is trained on sketches with a variable number of strokes, thanks to the arbitrary sum (i.e. $\sum_{\mathbf{s}\in \mathcal{S}}$) present in the true likelihood in Eq(3). Our SRM is conditioned on a latent vector, and at inference time, we can indeed generate any number of strokes. However, this is a parameter that has to be set manually. In Section 4.2 we discuss what happens when we vary the number of strokes during generation, and Figure 4 shows the results with varying numbers of strokes. For the results in the paper (unless specified otherwise), we generated 1000 strokes per sketch as we found it to produce good results. Since we assume that strokes are IID, we need to sample more strokes than the average cardinality of sketches in the training set (in Anime-Vec10k, there are 305 strokes per sketch on average). We discuss in detail in Section 4.2 oversampling and under-sampling.
>
> **Assumption of IID strokes:**
>
> We do not claim that strokes are independent — the strokes are indeed correlated in their true sense. However, we use the fact that the strokes are conditionally independent given a set representation (i.e. a compact latent code), powered by De-Finetti’s Theorem. The implication of this is that conditioned on a latent set representation, our SRM model will not result in a degenerate solution.
>
> **How is the sequence $\mathbf{s}$ generated at training time:**
>
> Thanks, but there might be a misunderstanding here. We provide below clarifications on our interpretation of the question —
> *“In the same way, how is the sequence generated at training time? Do you let the randomness of the random strokes take care of matching the proper final stroke? Why is there no Hungarian matching? Assuming normalized coordinates in [-1,1]x[1,1], how does it make sense for the denoising network to take a random stroke at (-1, -1) and try to denoise it to (1,1), instead of just recognizing (with Hungarian matching) or simple Euclidean distance that it is much easier to denoise the stroke closer to the final location?”.* Please do get back if this is incorrect — we will endeavor to respond promptly.
>
> First and foremost, we *do not* deal with sequence representations.  Each stroke is a 6-dimensional vector, as described in the beginning of Section 3.
> Also, we *do not* match strokes explicitly during training. During training, for each noise level, we pass to our MLP the noisy strokes (noisy control points, essentially) with added noise corresponding to the given noise level, and the latent code representing the set. Our MLP is simply trained to return a noise estimation per stroke, conditioned on the set representation. The simplicity of the SRM formulation is that it can simply be seen as training any other conditional diffusion model where the set latent is the condition and the strokes of one sketch as IID data points.
>
> **Numerical comparisons**
>
> Thanks. We have added a comparison of the FID scores of the generated drawings under different sampling conditions in Section 4.2 Table 1.
>
> **Additional references**
>
> Thanks for the additional references!
> We added a discussion of additional works to a revised introduction.
>
> **Additional qualitative results**
>
> Thanks for the suggestions! We have added additional qualitative results in Figure 7 and Figure 12 as well as examples of the training data in Figure 13.
>
> **Figure 5: Why does the DDPM sampler seem to have more strokes than the DDIM sampler? or are the strokes just placed closer together in the DDIM sample?**
>
> All samples generated in Figure 5 have the same number of generated strokes. Since DDPM sampling is stochastic, there is more diversity in stroke positioning. In DDIM sampling (which is deterministic), strokes are positioned so closely that it is hard to see that there are multiple strokes on top of each other. In Figure 15 in the Appendix, we reduce stroke width to better show the alignment between strokes for the DDIM sampling method.
>
> **Do you plan to release the dataset?**
>
> Yes, most definitely — this is a dataset that (we think) is much needed to progress the sketch community in terms of studying complex sketches. We will also release the code as well.

---

> > ### Comment · Reviewer_DN8s · 2023-11-23
> > **Response**
> >
> > Thank you for the updated manuscript and the response.
> >
> > 1. Sequence representation.
> >
> > Apologies here. I think I misunderstood in the initial questions that the representation is set based, so ordering is not even defined.
> >
> > This also leads into the iid question where I was confused as well.
> >
> > I think both would be clarified even more if you explicitly show how each denoising mlp is conditioned on the set representation. Eq 4 makes it clear, but Fig 14 is quite devoid of details
> >
> > This could include details liike how the set condition is applied - is it contacentated to the noise, added etc. we don’t know.
> >
> > Apart from that, the new manuscript is improved and I would raise my rating to a weak accept in the reviewer discussion phase.

---

> > > ### Comment · Reviewer_DN8s · 2023-11-23
> > > **Response**
> > >
> > > I would also like the authors to address the concern about over fitting.
> > >
> > > This is a valid concerns given the size of the dataset, and also the fact that the validation scheme is not described.

---

### Meta-Review · Area_Chair_dpDq · 2023-12-12

**Metareview:**

The paper deals with the difficult vector drawing generation problem and proposes a generative model based on Set Representation instead of the traditional sequence representation.
Strengths include the first method to generate complex drawings and the closed set representation.
Weaknesss include the concern on overfitting, the limited size of the training set (10K samples), poor visual quality, and lack of comparison (minor).
The paper should add more qualitative results (more noun classes), improve the visual quality, add comparison to previous methods, and discussion on limited size of dataset and potential overfitting.

**Justification For Why Not Higher Score:**

All reviewers gave borderline accept (note that one reviewer who gave borderline reject said he/she would increase the score but forgot to do so).
The reviewers appreciated the challenging nature of the problem but felt that the quality of the paper needs improvement. Thus, the paper does not achieve the next level.

**Justification For Why Not Lower Score:**

Two reviewers gave borderline accept.
One reivewer gave borderline reject. But based on the reviewer's comments, AC feels that the reviewer is inclined to borderline accept, quote "Apart from that, the new manuscript is improved and I would raise my rating to a weak accept in the reviewer discussion phase."

---

### Decision · Program_Chairs · 2024-01-16

Accept (poster)